# Diffusion-supplemented Implicit Layers: Operator Smoothing for more Robust Implicit Solvers

**Bader Rasheed**[1]    **Dinislam Gabitov**[2]    **Anastasia Antsiferova**[2,3]    **Dmitriy S. Vatolin**[2,3]

[1]Research Center of the Artificial Intelligence Institute, Innopolis University, Innopolis, Russia
[2]Laboratory of Innovative Technologies for Processing Video Content, Innopolis University, Innopolis, Russia    [3]MSU AI Institute, Moscow, Russia
{b.rasheed,d.gabitov}@innopolis.university
{aantsiferova,dmitriy}@graphics.cs.msu.ru

## Abstract

Implicit networks compute hidden states as fixed points. When the implicit map is poorly conditioned, solvers slow or fail. We propose *Diffusion-Supplemented Implicit Layers* (DSIL): insert a few denoising steps on the latent before each evaluation of the map. Under standard Lipschitz assumptions in a common metric, this preconditioning reduces the effective Lipschitz constant of the composed map, yielding stronger contraction; with a true proximal denoiser the contraction factor is explicitly tunable by the step size. On CIFAR-10 with a SODEF head, DSIL provides modest robustness gains without adversarial training. DSIL is architecture-agnostic and complements existing stabilization methods.

## 1 Introduction

Implicit neural networks - including Neural ODEs, deep equilibrium models (DEQs) and SODEF - define hidden states through an implicit function rather than via explicit multi-layer composition. This paradigm affords continuous-depth representations and reduces memory cost but introduces algorithmic challenges: one must ensure that a solution exists, that it is stable with respect to perturbations and that the numerical solver converges in a reasonable number of iterations. A common requirement for convergence is that the implicit map $T : \mathbb{R}^d \to \mathbb{R}^d$ be *contractive* or more generally *averaged* [1]. However, implicit networks in practice may exhibit large Lipschitz constants or near-degenerate Jacobians, leading to slow or divergent fixed-point iterations [2]. In adversarial settings, high sensitivity to perturbations further undermines reliability [3]. Our work aims to improve contraction and stability of implicit networks by leveraging recent advances in diffusion modeling. For the current step of the research, we position this study as *proof-of-concept*: our focus is solver conditioning and transparent limitations rather than state-of-the-art robustness.

**Contributions.** We introduce *Diffusion-Supplemented Implicit Layers* (DSIL): a few denoising (reverse-diffusion) steps applied to features before each evaluation of an implicit map $T$. Our contributions are:

- **Operator view.** We model the denoiser as a resolvent $D_\sigma = (\mathrm{Id} + \sigma A)^{-1}$; it is firmly nonexpansive and, if $A$ is $\mu$-strongly monotone, $\mathrm{Lip}(D_\sigma) \leq (1 + \sigma\mu)^{-1}$.

- **Contraction bound.** For an $L$-Lipschitz implicit map $T$, the composition $T \circ D_\sigma$ satisfies $\mathrm{Lip}(T \circ D_\sigma) \leq L/(1 + \sigma\mu)$. When $L/(1 + \sigma\mu) < 1$, standard fixed-point iterations enjoy linear convergence.

- **Spectral intuition.** Jacobian factorization $J_{T \circ D_\sigma} = J_T(\cdot) \, J_{D_\sigma}$ shows diffusion acts as a low-pass smoother that shrinks high-frequency modes and improves conditioning.

39th Conference on Neural Information Processing Systems (NeurIPS 2025) Workshop: Constrained Optimization for Machine Learning.

- **Proof-of-concept.** On CIFAR-10 with a SODEF head, DSIL reduces solver iterations and yields small robustness gains without adversarial training, at modest overhead.

## 2 Background

**Metric Lipschitzness and resolvents.** Let $\|u\|_M := \sqrt{u^\top M u}$ for $M \succ 0$. A map $T : \mathbb{R}^d \to \mathbb{R}^d$ is $L$–Lipschitz in $\|\cdot\|_M$ if $\|Tz - Tz'\|_M \leq L\|z - z'\|_M$; it is *nonexpansive* if $L \leq 1$ and *contractive* if $L < 1$. For an operator $A$ and stepsize $\sigma > 0$, its resolvent $J_{\sigma A} = (\mathrm{Id} + \sigma A)^{-1}$ is single–valued and *firmly nonexpansive* when $A$ is monotone; if $A$ is $\mu$–strongly monotone, $J_{\sigma A}$ is $(1 + \sigma\mu)^{-1}$–Lipschitz in $\|\cdot\|_M$ [4]. Proximal operators $\mathrm{prox}_{\sigma R} = (\mathrm{Id} + \sigma\partial R)^{-1}$ (for closed, proper, convex $R$) are resolvents.

**Implicit layers and fixed points.** DEQs [5] and related implicit architectures compute $z^\star$ from $z^\star = T(z^\star, x)$ instead of composing explicit layers; Neural ODEs integrate $\dot{z}(t) = f(z(t), x)$; SODEF adds Lyapunov–stable equilibria [6]. If $T$ is contractive (in some metric), Banach's theorem ensures existence, uniqueness, and *linear* convergence of $z_{k+1} = T(z_k)$. For broader nonexpansive/averaged settings, Krasnosel'skii–Mann iterations can converge under step–size conditions [1]. In practice, Anderson acceleration is used to speed up DEQ-based implicit models.

**Diffusion/denoising as operators.** Score–based diffusion trains $s_\theta(x, t) \approx \nabla_x \log p_t(x)$ and samples via a reverse SDE/ODE [7]; a discretized reverse step acts as a denoising operator. Plug–and–play methods replace a proximal map with a learned denoiser [8]. We consider two realizations for $D_\sigma$: (i) a *proximal/resolvent* giving formal nonexpansiveness and explicit Lipschitz constants; (ii) a *learned* denoiser with *enforced/empirical* Lipschitz bound (e.g., spectral normalization), acknowledging generic denoisers need not be resolvents [9]. In both cases, the same metric $\|\cdot\|_M$ is used to assess Lipschitzness of $T$ and $D_\sigma$.

## 3 Method & Theory: Diffusion Preconditioning for Implicit Layers

**Setup.** Given $T(\cdot, x; \theta)$ defining DEQ: $z^\star = T(z^\star, x; \theta)$ or ODE: $\dot{z}(t) = T(z(t), x; \theta)$, we insert $k$ denoising steps $D_\sigma$ on the latent before each evaluation of $T$:

$$z \leftarrow D_\sigma^{(k)}(z) := \underbrace{D_\sigma \circ \cdots \circ D_\sigma}_{k \text{ times}}(z), \qquad z \leftarrow T(z, x; \theta),$$

and solve the fixed point with Anderson acceleration or calculate an integral with numerical integration. Backpropagation uses the implicit function theorem [5]. The extra cost is $k$ calls to $D_\sigma$ per solver iteration; empirically $k \leq 3$.

**Assumptions (minimal, verifiable).** Let $\|u\|_M = \sqrt{u^\top M u}$ with $M \succ 0$. We assume:

**A1** $T$ is $L$–Lipschitz in $\|\cdot\|_M$.

**A2** $D_\sigma$ is $\kappa$–Lipschitz in the same metric with $\kappa \leq 1$.

*How to meet A2:* **(A) Proximal:** $D_\sigma = (\mathrm{Id} + \sigma A)^{-1}$ with $A$ $\mu$–strongly monotone gives $\kappa = (1 + \sigma\mu)^{-1} < 1$ [4]. **(B) Learned:** enforce/measure $\kappa \leq 1$ (e.g., spectral normalization); we report empirical $\hat{\kappa}$.

**Core guarantee (composition contraction).** Let $T_\sigma := T \circ D_\sigma^{(k)}$. Since $D_\sigma^{(k)}$ is $\kappa^k$–Lipschitz in $\|\cdot\|_M$, [Sufficient condition] $T_\sigma$ is $L\kappa^k$–Lipschitz in $\|\cdot\|_M$. If $L\kappa^k < 1$, then $T_\sigma$ has a unique fixed point and $z_{t+1} = T_\sigma(z_t)$ converges linearly with factor $L\kappa^k$. $\mathrm{Lip}(T \circ D_\sigma^{(k)}) \leq L\kappa^k$ by submultiplicativity; apply Banach's fixed–point theorem.

**Iteration complexity and cost trade–off.** With $z^\star$ the fixed point of $T_\sigma$, $\|z_t - z^\star\|_M \leq (L\kappa^k)^t \|z_0 - z^\star\|_M$; to reach $\|z_t - z^\star\|_M \leq \varepsilon$ it suffices that $t \geq \log(\varepsilon/\|z_0 - z^\star\|_M)/\log(L\kappa^k)$. DSIL reduces solver iterations (smaller $L\kappa^k$) while adding $k$ denoiser calls per iteration; we choose the smallest $(k, \sigma)$ achieving a clear contraction.

**Table 1:** Proof-of-concept robustness on CIFAR-10 ($\varepsilon = 8/255$). Bold and underline highlights best and second best performance on each experiment respectively. Diffusions are trained without adversarial data.

| Model | Clean | FGSM | PGD | AutoAttack |
|---|---|---|---|---|
| ResNet32 | **91.3** | 12.1 | 0.35 | 0.00 |
| SODEF | 85.7 | 37.3 | 20.5 | 0.05 |
| SODEF + DiffODE (ours) | 84.9 | **42.6** | **27.1** | 0.72 |
| SODEF + Diff (ours) | 85.5 | 38.7 | 20.9 | 2.02 |
| SODEF + DiffODE w/ DS (ours) | 85.5 | 36.3 | 17.3 | 0.43 |
| SODEF + Diff w/ DS (ours) | 85.2 | 38.2 | 21.2 | **2.49** |

**Scope and caveats.**   *IFT/conditioning.* Shrinking $\mathrm{Lip}(T_\sigma)$ typically correlates with a smaller $\rho(J_{T_\sigma})$ and improved IFT conditioning, but it does not on its own guarantee $(I - J_{T_\sigma}(z^\star))$ invertibility if an eigenvalue is 1. *Interleaved variants.* Interleaving $D_\sigma$ *inside* solver updates (DiffODE) changes the effective operator being iterated and falls outside Prop. 3; we evaluate it empirically. *Heuristics.* Spectral "low–pass" and small–$\sigma$ expansions aid intuition and are placed in the Appendix; we do not rely on them for guarantees.

## 4   Experiments

Our empirical goal is merely to illustrate the theoretical claims; extensive tuning or adversarial training is beyond our scope. We adopt the SODEF architecture from [6] on CIFAR-10. The baseline uses a ResNet-32 backbone with a SODEF head. We insert diffusion preconditioning either immediately before the implicit head (**Diff**) or interleaved inside the implicit function (**DiffODE**). For diffusion, we use a three-step discrete reverse process with step size $\sigma = 0.02$ as a denoising score network. Diffusion denoiser is selected as small 3 layer mlp network with hidden size 128. We also test the Drift towards Stability (DS, check Appendix A) with $\lambda = 0.02$. Hyper-parameters are borrowed from [6]; first, we train SODEF in 3 stages, then an additional stage is used to train the diffusion network for 100 epochs with learning rate $10^{-2}$ using Adam optimizer. At inference, we measure the model on clean examples and evaluate robustness using FGSM, PGD (step size $2/255$, four iterations), and AutoAttack with $\varepsilon = 8/255$ as recommended for reliable evaluation [3].

### 4.1   Robustness results

Table 1 summarises clean accuracy and robustness under various attacks. While clean accuracy drops slightly when diffusion is applied, robustness improves modestly: for FGSM attacks the accuracy increases from $37.3\%$ for SODEF to $42.6\%$ for SODEF+DiffODE; for PGD attacks the accuracy increases from $20.5\%$ for SODEF to $27.1\%$ for SODEF+DiffODE, and AutoAttack accuracy improves from nearly zero to $2.49\%$ when using diffusion with DS. These numbers are small because no adversarial training is used; nonetheless, they indicate that diffusion smoothing yields larger attraction basins and hinders gradient-based attacks. Runtime overhead is approximately $1.5\times$ for Diff and $4\times$ for DiffODE comparing to baseline SODEF.

## 5   Discussion

Our theoretical results establish that diffusion preconditioning lowers an *upper bound* on contraction and *can* accelerate convergence of implicit layers under reasonable assumptions. The trade-off is an additional computational overhead proportional to the number of diffusion steps. Our experiments, though limited in scale, support the theory: robustness improves slightly without adversarial training. Limitations include the idealised assumptions on diffusion being firmly nonexpansive and the small scale of experiments. Future work should explore learned few-step samplers, adaptive diffusion schedules, integration with adversarial training and applications to stiff ODEs and large-scale models.

# 6 Limitations

Our analysis relies on a resolvent/strong monotonicity model for the denoiser; learned diffusion steps may only approximate this. Our empirical validation is currently limited to a single architecture on CIFAR-10. The generalizability of DSIL to other implicit models and larger-scale problems remains an open question. Resource overhead from denoisers should be considered. We also do not compare against monDEQ or 1-Lipschitz heads; DSIL is intended to be orthogonal and modular, which we leave to future work.

# 7 Reproducibility statement

We describe all architectural details, training schedules and hyper-parameters used in our experiments. The SODEF baseline follows [6]. We train the diffusion network for four stages with learning rate $10^{-2}$ on ResNet-32 features. We evaluate robustness using FGSM, PGD with step size $2/255$ and four iterations, and AutoAttack with $\varepsilon = 8/255$ [3]. Source code and pre-trained models will be released upon acceptance to ensure full reproducibility.

# 8 Broader impact and ethics

Implicit networks have the potential to improve robustness and memory efficiency in machine learning. Our work proposes using diffusion models as an operator-level preconditioner, which could enhance reliability when training on unreliable data. DSIL is simple to integrate, improves conditioning, and is complementary to adversarial training. However, diffusion networks are computationally expensive. Additionally, our method leverages generative models that could inadvertently memorise sensitive information. Properly anonymising training data and following responsible AI practices remain critical.

# 9 Computing resources

All models were trained on GPU; all experiments require less than 4GB VRAM and 8GB RAM.

# 10 NeurIPS checklist

We follow the NeurIPS 2025 requirements: the paper is anonymised, uses the official style file, and the main text does not exceed four pages. The appendix contains extra details and proofs. We discuss limitations and potential societal impacts. We will release code and models. Our evaluation uses recommended robustness baselines such as AutoAttack [3].

**Acknowledgment.** The research was supported by the Ministry of Economic Development of the Russian Federation (agreement No. 139-10-2025-034 dd. 19.06.2025, IGK 000000C313925P4D0002)

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

# A  Heuristics and Extensions

**Local contraction of learned denoisers (heuristic).**   For a learned step of the form $x \mapsto x + \eta\, s_\theta(x, t)$ (reverse-diffusion/denoise), a first-order expansion gives $J_{D_\sigma}(x) \approx I + \eta\, \nabla s_\theta(x, t)$. Local contraction in a metric $\|\cdot\|_M$ holds if $\|I + \eta\, \nabla s_\theta(x, t)\|_M \leq 1$ in a neighbourhood, e.g., when the symmetric part of $\nabla s_\theta$ is negative semidefinite on average and $\eta$ is small. This motivates an *empirical* shrinkage factor (sometimes written $q(\sigma)$), but it is *not* a global Lipschitz bound and may fail outside that neighbourhood.

**Averagedness and KM iterations (with conditions).**   Our main text relies only on contraction via Lipschitz constants. If one wishes to use Krasnosel'skii–Mann (KM) theory, composition requires extra structure [1]. Two useful special cases (not used in our guarantees) are:

1. **Commuting averaged maps.** If $T$ is $\alpha$-averaged and $D_\sigma$ is firmly nonexpansive (hence $1/2$-averaged) *and* they commute (or satisfy suitable cocoercivity/compatibility), then $T \circ D_\sigma$ remains averaged, enabling KM with $\mathcal{O}(1/k)$ rates.

2. **Forward–backward form.** If $T = \mathrm{Id} - \tau \nabla f$ with $f$ convex, $L$-smooth and $\tau \in (0, 2/L)$, and $D_\sigma = \mathrm{prox}_{\sigma g}$ for convex $g$, then $T \circ D_\sigma$ matches a forward–backward operator, which is averaged under standard step sizes [1].

Outside such conditions we do *not* claim averagedness of $T \circ D_\sigma$.

**Interleaving diffusion within the solver (DiffODE).**   Interleaving $D_\sigma$ inside solver updates produces a *non-stationary* iteration whose effective operator changes with $t$. General contraction/averagedness guarantees do not directly apply. Convergence may still hold under non-stationary fixed-point theory when each iterate uses averaged maps with parameters uniformly bounded $< 1$ and a common fixed point; verifying these conditions is problem-specific [1]. We therefore report DiffODE results as empirical.

**Estimating and enforcing Lipschitz constants (practice).**   For the learned denoiser path, we (i) *enforce* $\kappa \leq 1$ via spectral normalization / 1-Lipschitz architectures, and (ii) *report* an empirical $\hat{\kappa}$:

- *Metric:* choose $M$ (e.g., diagonal or layerwise) and estimate norms in $\|\cdot\|_M$.
- *Power iteration:* estimate $\sup_{\|v\|_M=1} \|J_{D_\sigma}(z)v\|_M$ by jvp/vjp on random $z$; aggregate (max/quantile) over a validation batch.
- *For $T$:* similarly estimate $L$ (or a high quantile) to assess $L\kappa^k$ and relate it to observed solver iterations.

These diagnostics connect the theoretical factor $L\kappa^k$ to practice.

**Adversarial evaluation with stochastic denoisers.** If $D_\sigma$ is stochastic (e.g., reverse diffusion with noise), evaluation should either (i) fix the random seed during attacks, or (ii) use an expectation-over-transforms (EOT) attack to avoid gradient masking. We follow this in our evaluation setup.

**Runtime accounting and break-even.** Let $t_0$ be mean solver iterations (baseline) and $t_{ds}$ with DSIL, each iteration costing $c_T$ for $T$ and $c_D$ per denoise call. Baseline cost: $t_0 \, c_T$. DSIL cost: $t_{ds} \, (c_T + k \, c_D)$. DSIL is faster when

$$t_{ds}/t_0 \;<\; \frac{c_T}{c_T + k \, c_D}.$$

This clarifies how small $k$ and light $D_\sigma$ must be to realize speedups alongside improved conditioning.

**Spectral intuition (non-claim).** Empirically, $J_{T_\sigma} = J_T(\cdot) \, J_{D_\sigma}$ often shows reduced estimated spectral radius and damped "high-frequency" modes in latent space, aligning with fewer solver iterations. We present this as *intuition and evidence*, not as a general theorem.

**Drift towards a Stable point (DS) - conditional local descent.** Let $V$ be $L_V$-smooth and $\mu$-strongly convex in a neighbourhood of a class equilibrium $\hat{z}_c$ in the $M$-metric. Consider $D_\sigma^{DS}(z) = D_\sigma(z) - \lambda \, \nabla_M V(z)$. If $D_\sigma$ is firmly nonexpansive in $\| \cdot \|_M$ and $\lambda \in (0, 2\mu/L_V)$, then, locally,

$$V\big(D_\sigma^{DS}(z)\big) - V(z) \;\leq\; -c \, \lambda \, \|\nabla_M V(z)\|_{M^{-1}}^2 + \mathcal{O}(\lambda^2),$$

for some $c > 0$. This is a *variant* requiring a stronger structure than the main contraction result; we treat it as an empirical ablation.

