# OpenReview forum: "Diffusion-supplemented Implicit Layers: Operator Smoothing for better Implicit Solvers"
_NeurIPS.cc/2025/Workshop/Reliable_ML — NeurIPS 2025 - Reliable ML Workshop_

### Official Review · Reviewer_Xmr2 · 2025-09-16
**Using diffusion steps to condition implicit solvers.**

**Rating:** 6
**Confidence:** 2

**Review:**

The authors proposed to use a several diffusion steps to condition implicit solvers. The steps effectively behave like a low pass filter.

The idea seems simple and working in the proof of concept experiments.

I am not fully sure that how diffusion, in this case, is different from having low pass filter and appears to me that the algorithm is indeed solving a linear system.

I would suggest the author to make a comment on how this is related to the diffusion literature more explicitly.

---

### Official Review · Reviewer_WNYY · 2025-09-20
**Diffusion-supplemented Implicit Layers: Operator Smoothing for better Implicit Solvers**

**Rating:** 5
**Confidence:** 2

**Review:**

Summary: They propose Diffusion-Supplemented Implicit Layers (DSIL). Idea: before each implicit map step (e.g., DEQ/SODEF), run a tiny denoise/diffusion on the latent, so the map becomes more contractive and the solver converges easier. They argue the denoiser acts like a resolvent (shrinks effective Lipschitz), give some bounds, and show small robustness gains on CIFAR-10 with a SODEF head (PGD/FGSM) — but it costs extra time.

Strengths: Theoretical results of the paper look good(operator view, contraction story makes sense). Easy plug-in (architecture-agnostic). Experiments match the claim: a bit fewer iterations / a bit more robustness without doing full adversarial training.

Weaknesses: Presentation could be cleaner — esp. the experiments: more runs, clearer charts/bars, and error bars would help. Evidence is narrow (single dataset/arch), and some assumptions (denoiser ≈ nonexpansive/resolvent, Lipschitz/monotone) feel stronger than needed or only approximate in practice. Runtime overhead isn’t small (~1.5×–4×), and compute/variance isn’t well reported.

Suggestions: look at weaknesses.